# CLUSTERING MEETS IMPLICIT GENERATIVE MODELS

**Francesco Locatello**[1,3], **Damien Vincent**[2], **Ilya Tolstikhin**[1], **Gunnar Rätsch**[3], **Sylvain Gelly**[2], **Bernhard Schölkopf**[1]

[1] Max Planck Institute for Intelligent Systems, Tübingen, Germany
[2] Google Brain
[3] ETH Zurich, Switzerland
[1]`{flocatello,ilya,bs}@tuebingen.mpg.de`
[2]`{damienv,sylvaingelly}@google.com`
[3]`{raetsch}@inf.ethz.ch`

## ABSTRACT

Clustering is a cornerstone of unsupervised learning which can be thought as disentangling multiple generative mechanisms underlying the data. In this paper we introduce an algorithmic framework to train mixtures of implicit generative models which we particularize for variational autoencoders. Relying on an additional set of discriminators, we propose a competitive procedure in which the models only need to approximate the portion of the data distribution from which they can produce realistic samples. As a byproduct, each model is simpler to train, and a clustering interpretation arises naturally from the partitioning of the training points among the models. We empirically show that our approach splits the training distribution in a reasonable way and increases the quality of the generated samples.

## 1 INTRODUCTION

In recent years, (implicit) generative models have attracted significant attention in machine learning. Two of the most prominent approaches are Generative Adversarial Networks (GANs) (Goodfellow et al., 2014) and Variational Autoencoders (VAEs) (Kingma & Welling, 2013). Both approaches aim at minimizing the discrepancy between the true data distribution and the one learned by the model. The model distribution is typically parametrized with a neural network which transforms random vectors into samples in the space of the training data (e.g., images). Variational Autencoders maximize a log-likelihood and are able to perform efficient approximate inference on probabilistic models with continuous latent variables and intractable posterior. Unfortunately, VAEs are known to produce blurry samples when applied to natural images. GANs take a completely different approach, relying on adversarial training. This resulted in impressive empirical results. On the other hand, adversarial training comes at a cost. GANs are harder to train and suffer from the *mode collapse* problem. One solution to this problem is to train multiple generative models either sequentially (Tolstikhin et al., 2017) or in parallel (Hoang et al., 2017). In contrast to GANs, VAEs suffer huge loss if they do not model the whole support of the data distribution with sufficiently high probability. As a consequence, they often place a significant probability mass in regions outside the support of the data distribution.

We aim at bridging this gap, developing a general approach to train multiple generative models in parallel which focus on different parts of the training distribution. We particularize this framework for VAEs. As a consequence, each VAE will be able to collapse on some modes while the mixture of generators (decoders) will still approximate the whole data distribution. Borrowing ideas both from the literature on clustering and on causality, we assume that the data was generated by *independent mechanisms*, i.e., that the generative process of the overall distribution is composed of separate modules that do not inform nor influence each other (Peters et al., 2017). Consider the special case of a variable $X_0$ which is caused by (mixing) several independent sources $X_1, \ldots, X_K$ without parents in the causal graph. In this case, the causal generative model can be written as

$$p(X_0, \ldots, X_K) = p(X_0|X_1, \ldots, X_K) \prod_{j=1}^{K} p(X_j). \tag{1}$$

Note that only one of the mechanisms, $p(X_0|X_1, \ldots, X_K)$, implementing the mixing, is still a conditional; the others reduce to unconditional distributions since the sources have no parents. The conditional can be written as a structural equation (Pearl, 2000) in the form of $X_0 := f(X_1, \ldots, X_K, j) \equiv X_j$, where $j$ is a discrete noise variable taking values in $\{1, \ldots, K\}$. The distribution of $j$ determines the mixing coefficients. This structural equation expresses the conditional as a mechanism represented by a noisy function. Suppose each training point was generated by one of the mechanisms $X_j$, but we get to observe only the mixture $X_0$ of all these realizations. Recovering the mechanisms would amount to learning a particular kind of structural causal generative models, and it could form a building block of more complex causal models (Schölkopf et al., 2016). We make the simplifying assumption that the supports of the different mechanisms do not overlap, hence if we observe two identical realizations of $X_0$ we assume they were generated by the same mechanism.

## 2 TRAINING INDEPENDENT GENERATIVE MODELS

---

**Algorithm 1** Mixture training

---

1: **init** $K$ generative models $g_j$, $c_j^{(0)}$
2: **for** $t = 0 \ldots T$
3: $\quad \min_{P_{g_j}} D_f \left( P_{g_j} \| P_{d_j}^{(t)} \right)$ for every $g_j$ in parallel.
4: $\quad$ Update $c_j^{(t+1)}(x)$ for every $g_j$
5: **end for**

---

Let $\mathbf{X}$ be a dataset composed of $N$ samples $x$ from the data space $\mathcal{X}$, which are realizations of $X_0$. Furthermore, let $P_d$ be an unknown data distribution defined over the data space $\mathcal{X}$ with support $\mathsf{X}$ to be approximated with an easy to sample distribution $P_{model} = \sum_{j=1}^{k} \alpha_j P_{g_j}$ by minimizing an $f$-Divergence (Nowozin et al., 2016). Each component $P_{g_j}$ should specialize on one of the generating mechanisms. Intuitively, our training procedure is related to the k-means algorithm. In k-means, one first decouples the training data across the centroids and then update the centroid based on the assignment. Our approach is informally depicted in Algorithm 1. For a given assignment function $c_j$ we define $dP_{d_j}$ as the density obtain by normalizing $dP_d(x)c_j(x)$. We further define the weighting $\alpha_j$ as the normalization constant of $dP_{d_j}$. This can be empirically estimated by counting how many training points are assigned to the $j$-th generator. We can decouple the training of the generators by minimizing an upper bound of the $f$-divergence (proof in the Appendix):

$$\min_{P_{g_j}} \sum_j \alpha_j D_f(P_{g_j}|P_{d_j}), \tag{2}$$

Since each term in the sum in Equation (3) is independent, each generative model can be trained independently to approximate $dP_{d_j}^{(t)}$.

After training the generative models, we fix them and update the assignment of each training point by maximizing an estimate of their likelihood. We train a discriminator to distinguish samples from $P_{g_j}$ and samples from $P_d$, for which we have that:

$$dP_{g_j}^{(t)}(x_i) \approx dP_d(x_i) \frac{1 - D_{g_j}^{(t)}(x_i)}{D_{g_j}^{(t)}(x_i)}$$

Note that this approximation makes sense only when computed on the training points. Therefore, we approximate $dP_{g_j}^{(t)}(x_i)$ as the empirical estimate over the training set. We now assign each training point to the mechanism $j$ that generates the most similar samples, i.e. $c_j^{(t+1)}(x_i) = 1$ if $j = \arg\max_l dP_{g_l}^{(t)}(x_i)$ and 0 otherwise. A sketch of the competitive training procedure, using VAEs decoders as generators (which we call kVAEs), is depicted in Figure 1. In the Appendix we further discuss the clustering interpretation.

## 3 EXPERIMENTAL PROOF OF CONCEPT

In Figure 2 we depict the output of our algorithm trained on synthetic data from a distribution with 5 different modes, in which one is more complex than the others. In Table 1a we report the log likelihood of the true data under the generative model distribution.

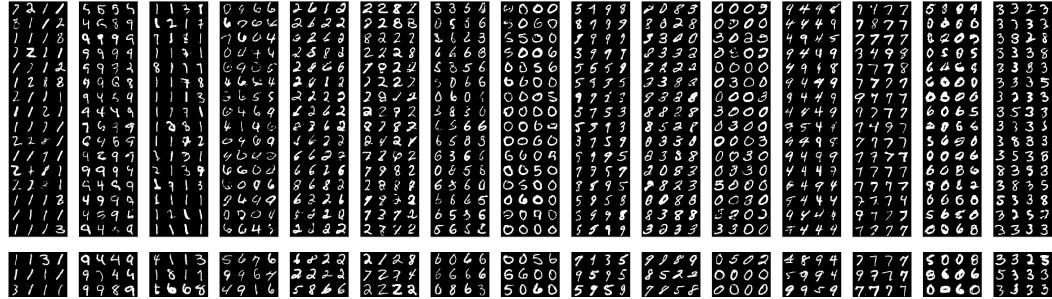

Figure 3: MNIST: samples generated by 15 mixture components and real digits clustered after 10 iterations

For the experiment on MNIST, we do not know the number of modes. There is no reason to believe the optimal number of modes should be the number of digits. We arbitrarily use 15 models, to also capture stylistic differences between digits, following the insights from (Tolstikhin et al., 2017). Note that the different VAEs did specialize on distinct parts of the data distribution as similar digits and styles tend to be grouped together. In Table 1b we report the FID score (Heusel et al., 2017) and compare it against an ensemble of 15 VAEs trained uniformly and with bagging, a single VAE of the same capacity of the ones in the ensembles (maximum 64 filters per layer) and a larger VAE (maximum 512 filters per layer). We note that our approach increases the FID score of a single model and is competitive with larger models. At the same time, our approach is more scalable, as it allows one to decouple the complexity of the overall model across the different VAEs. All experimental details are deferred to the Appendix.

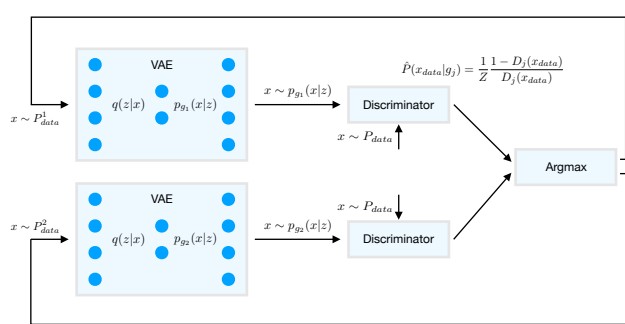

Figure 1: Training pipeline

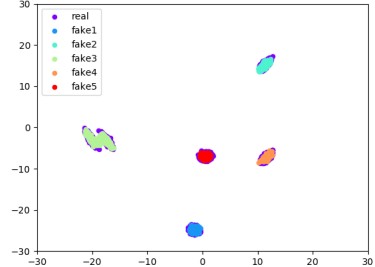

Figure 2: Synthetic data experiment, 5 modes and 5 VAEs (initialized with 100 epochs of uniform training) after 130 iterations of the meta-algorithm.

| experiment | kVAEs | bag | VAE-150 |
|---|---|---|---|
| 3 modes | **-4.59** | -6.49 | -5.42 |
| 5 modes | **-2.74** | -7.7 | -5.71 |
| 9 modes | **-2.51** | -7.05 | -6.83 |

(a)

| kVAEs | kVAEs init | bag | VAE-64 | VAE-512 |
|---|---|---|---|---|
| 9.99 | 15.33 | 21.38 | 17.96 | 9.44 |

(b)

Table 1: **(a)** Log-likelihood of the true data under the generated distribution after 100 iterations of the kVAE algorithm with as many components as modes and 50 units per layer, 100 epochs of bagging and 1000 epochs of a single larger VAE (150 units per layer). **(b)** FID score on MNIST. A random split of the training set hurts the performance of the models as it does not carry any semantic and each model is trained on less data overall.

## 4  CONCLUSIONS:

In this paper, we introduced a clustering procedure using implicit generative models, which encourages them to generate more realistic samples. We train networks with limited capacity and let them compete between each other in the pursuit of generating more realistic samples. We empirically validated that the model can successfully recover the true generative mechanisms and in general allows one to generate samples which are closer to the support of the data distribution. In MNIST we obtained FID scores which are competitive with the ones of larger VAEs. The approach we presented is extremely modular and there are several possible extension. For example, given enough computational resources one can dramatically increase the number of generative models.

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

## A   Implementation Details

We generate synthetic data in 2 dimensions by first sampling 64,000 points from a Gaussian distribution and then we skew the second dimension $x_2$ with the non-linear transformation $x_2 = x_2 + 0.04 \cdot x_1^2 - 100 \cdot 0.04$. This ensures that each mode is sufficiently complex so that the VAEs cannot perfectly learn the data distribution by encoding each mode in a different dimension of the latent space. We use a small and standard architecture for the VAE: a neural network with two hidden layers with 50 units each as both the decoder and the encoder. The discriminator has a similar architecture. We use a 5 dimensional latent space and assume a Gaussian encoder. At each iteration, we train each VAE for 10 epochs on a split of the dataset (VAEs are pretrained uniformly on the dataset), and the classifier is trained for 2 epochs for the first two experiments and 5 for the third. We use Adam (Kingma & Ba, 2014) with step size 0.005, $\beta = 0.5$ batch size 32. We perform three different experiments.

We compute the log-likelihood of the true data under the generated distribution using a Kernel Density Estimation with Gaussian kernel. We compare against a larger VAE with 150 units per layer (instead of 50), trained uniformly over the training set, and a bag of VAEs with 50 units trained on a random subsample of the training set (sampled with replacement) containing $N/K$ training points. We note that the random splitting of the training set did not help the VAE to specialize and actually made the log-likelihood worse after training for 100 epochs.

### A.1   MNIST

We again use a small and simple architecture, with *relu* activation functions. The encoders and the decoders have 4 convolutional layers with 8-16-32-64 $4 \times 4$ filters. We use batch normalization with $\epsilon = 10^{-5}$ and decay 0.9. Each VAE has a cross-entropy reconstruction loss and a latent space dimension of 8, and we fix the learning rate of Adam for all networks to 0.005. The discriminator has 3 convolutional layers and a linear layer with number of filters 64-128-256. As opposed to the synthetic data example, we do not reinitialize the classifier at each iteration of the meta algorithm. Instead, we train it for a single batch in every iteration of the meta-algorithm. The reason is that we found the classifier output to be too sensitive to the initialization if it is not trained sufficiently long. On the other hand, training a full discriminator in every iteration was too expensive, and if trained too much, it would learn to distinguish fake example by just looking at specific blurriness patterns. In the synthetic experiments, the data produced by each VAE was indistinguishable from the real data if the support was correct, so training a classifier from scratch was feasible and gave best results.

To evaluate our generated samples we used the FID score (Heusel et al., 2017). We remark that our FID score is competitive with the one obtained with a large VAE, as well as the one that can be obtained with GANs (slightly less than 10 was reported in  (Lucic et al., 2017, Figure 5)).

## B   Proof of Equation 3

**Lemma 1.** *For a fixed partition function $c_j$, we minimize for all $j \in [K]$:*

$$\min_{P_{g_j}} \sum_j \alpha_j D_f(P_{g_j} \| P_{d_j}), \tag{3}$$

*which is an upper bound on the $f$-divergence for a mixture model.*

*Proof.* By definition of the model we write the $f$-divergence as:

$$D_f(P_{model} \| P_d) = D_f(\sum_{j=1}^k \alpha_j P_{g_j} \| P_d)$$

Now, we have that $\alpha_j = \int_{\mathsf{X}} dP_d(x) c_j(x)$. Since $\mathsf{X}^j \cap \mathsf{X}^k = \emptyset$ for $j \neq k$, we can write:

$$D_f(\sum_{j=1}^k \alpha_j P_{g_j} \| P_d) = D_f(\sum_{j=1}^k \alpha_j P_{g_j} \| \sum_{j=1}^k \alpha_j P_{d_j})$$

Joint convexity of $D_f$ concludes the proof. □

## C  CLUSTERING INTERPRETATION

The general approach we introduced is closely related to clustering. In this section we revisit classical clustering notions in view of our framework. We show that we generalize k-means in non-metric spaces, and we recover it when the space is euclidean.

In the generative interpretation of clustering, one assumes that the data was generated from each centroid $\mu$ with an additive Gaussian noise vector, i.e., $x = \mu + \epsilon$. This formulation naturally yields an euclidean cost for the cluster assignment when decoupling the data between the different centroids.

Unfortunately, the euclidean distance is known not to be a good metric for clustering images. Our goal in the present paper is to find a clustering of the data across the generating mechanism in a setting in which the metric of the space is not known.

We now show how to recover k-means clustering from our framework. Assume that the data is generated by a mixture of Gaussians. We can lower bound the log-likelihood of the data using a variational bound:

$$\log(P(X)) \geq \sum_i \sum_j q_i(j) \log \left( \frac{P(x_i, j)}{q_i(j)} \right)$$

where $q$ is the variational approximation of the posterior and $j$ is the index of the components. One can then simply rewrite $P(x_i, j) = P(x_i|j)p(j)$. Then, for a Gaussian mixture model one parametrizes $P(x_i|j)$ with a Gaussian distribution. If the Gaussian is isotropic and when the covariance vanishes one obtains that $q_i(j)$, the variational approximation of the posterior, degenerates to a hard assignment. Instead of approximating the generative model with a Gaussian distribution, we parametrize $P(x_i|j)$ with an implicit generative model from which it is easy to sample. If $P_j$ is the decoder of a variational autoencoder, we can obtain the k-means algorithm by assuming that the mean of the encoder is constant and independent of $x$. Note that VAEs are trained to maximize the log-likelihood as in EM. Assume we have a Gaussian encoder which maps all the input to a single point (degenerate Gaussian with $\sigma = 0$). Now, say we have the identity as decoder. The result of this procedure is that when training the autencoder with the reparametrization trick, one has to minimize

$$\mathbb{E}_{x \sim P_{d_j}} \left[ -\log P_{g_j}(x|\mu_j) \right] = \mathbb{E}_{x \sim P_{d_j}} \left[ \frac{1}{2} \|x - \mu_j\|^2 \right].$$

Then, using EM, we compute the update for the (degenerate) variational distribution:

$$q_i(j = 1) = \lim_{\sigma \to 0} \frac{\alpha e^{-\|x_i - \mu_j\|/2\sigma}}{\sum_j \alpha_j e^{-\|x_i - \mu_j\|/2\sigma}}$$

And recalling that $\log P_{g_j} = -\|x_i - \mu_j\|^2/2$ we notice that the degenerate posterior is obtained by maximizing the likelihood. Instead of using the autencoder loss, we estimate $P_{g_j}$ using a discriminator to account for the fact that we might not have a clear notion of distance. In an euclidean space, one could simply use a nearest neighbor classifier between the output of the VAEs (i.e. the centroids) and the training points. Note that this procedure is exactly k-means.

