# OpenReview forum: "Clustering Meets Implicit Generative Models"
_ICLR.cc/2018/Workshop — Accept_

### Official Review · AnonReviewer2 · 2018-03-09
**Learn a variety generative models to specialize on different subspaces of the data distribution**

**Rating:** 7
**Confidence:** 3

**Review:**

This paper aims to give a model composed of submodels (each of which is a VAE whose role is to focus on different parts of the training distribution) and a learning algorithm for it. The learning algorithm is motivated by the k-means algorithm. For each assignments of datapoints to submodels, maximize the likelihood of datapoints under each submodel (in parallel), then fix the generative submodels and train a discriminator to distinguish between the data distribution and the samples from each submodel. Reassign points to submodels that generate the most similar points (using an argmax over the discriminator's probability distribution) and repeat.

Pros:
* Interesting (and to my knowledge novel) idea
* The experiments are preliminary and synthetic though they appear to show the model working

Cons:
* Missing a discussion of (Boosted Generative Models) https://arxiv.org/pdf/1702.08484.pdf and a discussion of sensitivity of the learning algorithm to initialization
* A more natural baseline to compare against than a VAE might be a generative model that assumes a mixture distribution in the prior (see for example - https://arxiv.org/pdf/1603.06277.pdf).

Questions for the authors: How is c_j initialized? How sensitive is the learning algorithm to this quantity? Are the VAEs reset after each assignment of the c_js?

---

### Official Review · AnonReviewer1 · 2018-03-10
**Interesting work**

**Rating:** 6
**Confidence:** 3

**Review:**

This paper proposes an algorithmic framework to train mixtures of implicit generative models. The framework mimics the k-means algorithm, and the training can be decoupled into independent training each generative model separately through minimizing an upper bound.

The proposed method is very similar to that of [1], where a mixture of generative models are trained using a similar EM-style algorithm. The major difference is if one data point is "hard" or "soft"-assigned to one of the generative models. My guess is that "hard" version would train faster and generate more vivid images than the "soft" counterpart. But it would be interesting to see the quantitative comparisons.

- Please be consistent with the symbols: Line 3 in Algorithm 1 uses "D(||)", but equation (2) uses "D(|)".
- My computation shows that the number of parameters in VAE-150 is smaller than that of 9VAEs. Another issue is that if the number of parameters in the discriminator should also be counted. It would be interesting to see the performance of a single VAE with one more layer but similar number of parameters.
- It would be interesting to see the sensitivity of training with respect to the optimality/epochs of the discriminator.

[1] E. Banijamali, A. Ghodsi, and P. Poupart. Generative Mixture of Networks. https://arxiv.org/abs/1702.03307

---

### Official Review · AnonReviewer3 · 2018-03-11
**Interesting idea, poorly explained**

**Rating:** 4
**Confidence:** 4

**Review:**

As I understand it, this paper presents a mixture model p(x) = \sum_{k} p(x | class_k) p(class_k), where each p(x | class_k) is trained adversarially. The model is trained in an iterative way, much like k-means, where we alternate between computing class/cluster assignments and updating the individual mixture models p(x | class_k). The step calculating the cluster assignments is not probabilistic -- one seems to use a hard argmax to make hard cluster assignments, which produces a hard partition of the data, from which one can adversarially retrain k VAE-based models. I did like the idea of having an adversarial way to train the decoder part of the VAE, although I got confused when the authors said there was a "cross-entropy" reconstruction error (?). This leads me nicely to my main criticism of the paper -- it is unnecessarily confusing and written in a highly non-rigorous full of inconsistent notation and confusing statements.

The introduction talks about causality/structural equations (why?) and introduces X1...XK which are never really used after. All this wasted space could have been used to make a clearer/much more rigorous description of the main algorithm. For example, what is the difference between dP_{d_{j}} and P_{d_{j}}. What is D_{g_{j}}? and so on.

Empirical results are also weak. MNIST alone won't suffice in my opinion, you need at least SVHN. The FID Score is never defined -- if one is to use this frequently in a paper it would be nice to have a one-liner. Also, for the synthetic data experiment why not numerically compute the f-divergence, instead of the loglikelihood of the true data?

---

### Decision · Program_Chairs · 2018-03-20
**ICLR 2018 Workshop Acceptance Decision**

**Decision:**

Accept

**Comment:**

Congratulations, your paper was accepted to the ICLR workshop.